behaviour

ant cue, spider deterrent, integrated pest management, *Myrmica rubra*

**Author for correspondence:**
Andreas Fischer
e-mail: afischer@sfu.ca

†Present address: Department of Ecology and Evolutionary Biology, University of Toronto, Toronto, Ontario, Canada.
‡These authors contributed equally to this study.

# Know your foe: synanthropic spiders are deterred by semiochemicals of European fire ants

Andreas Fischer, Yerin Lee†,‡, T'ea Dong‡ and Gerhard Gries

Department of Biological Sciences, Simon Fraser University, Vancouver, British Columbia, Canada

AF, 0000-0001-5922-8411; GG, 0000-0003-3115-8989

Many ants prey on spiders, suggesting that web-building spiders may avoid micro-locations near ant colonies or frequented by foraging ants. Here we tested the hypothesis that ant-derived semiochemicals deter synanthropic spiders. To generate stimuli, we exposed filter paper for 12 h to workers of European fire ants, *Myrmica rubra*, black garden ants, *Lasius niger*, or western carpenter ants, *Camponotus modoc*, and then offered select urban spiders in three-chamber olfactometer bioassays a choice between ant-exposed filter paper and unexposed control filter paper. Semiochemical deposits of *M. rubra*, but not of *L. niger* or *C. modoc*, had a significant deterrent effect on subadults of the false black widow, *Steatoda grossa*, the black widow, *Latrodectus hesperus*, and the hobo spider, *Eratigena agrestis*, as well as a moderate (but statistically not significant) deterrent effect on the cross spider, *Araneus diadematus*. The deterrent effect caused by semiochemical deposits of *M. rubra* may be attributable to the aggressive nature and efficient foraging of *M. rubra* in its invaded North American range, exerting selection pressure on community members to recognize *M. rubra* semiochemicals and to avoid micro-locations occupied by *M. rubra*.

## 1. Introduction

Widespread arachnophobia [1,2] is fuelled, in part, by fear of the few neurotoxic spiders [3,4]. This fear has inspired the development of tactics to physically and chemically discourage synanthropic spiders from settling in and around human dwellings [5]. Proposed physical tactics include sealing holes and cracks in building walls, removing webs, reducing moisture and changing exterior lighting that attracts insect prey for

spiders [3]. Chemical tactics such as insecticide applications [5] are largely ineffective because spiders can avoid insecticides by abandoning their web and rebuilding one elsewhere [3,6]. Natural repellents of spiders, such as chestnuts and lemon oil, are widely advertised in anecdotal accounts but only a few have been experimentally tested [7,8], and none effectively repelled all species of spiders tested [9]. Moreover, there is no immediate ecological reason why these materials are repellent to spiders.

By contrast, there is every reason for spiders to avoid natural predators such as ants that prey on both web-building and cursorial spiders [10–12]. At the population level, there is a negative correlation between the density of ant populations and the total biomass of spiders [13,14]. Cobweb spiders, *Phylloneta impressa*, tend to disperse in response to chemical cues derived from black garden ants, *Lasius niger*, and the formicine ant *Formica clara* [15]. Sensing chemical cues of potentially predatory ants is particularly adaptive for subadult web-building spiders which seek suitable micro-locations for settling and building their webs [16]. As web building is a significant time and energy investment [17–19], subadult spiders are thought to explore, and ultimately select, primarily those microhabitats that have no or few threats to survival, such as the presence of predatory ants. Flat rock spiders, *Morebilus plagusius*, e.g. avoid ant-scented rocks when selecting retreat sites [20].

Here we tested the hypothesis that ant-derived semiochemicals deter spiders. As model organisms for our study, we selected three synanthropic ant species [European fire ants, *Myrmica rubra*; black garden ants, *Lasius niger*; western carpenter ants, *Camponotus modoc* (all Formicidae)] and four synanthropic web-building spider species [false black widow, *Steatoda grossa*; western black widow, *Latrodectus hesperus* (both Theridiidae); cross spider, *Araneus diadematus* (Araneidae); hobo spider, *Eratigena agrestis* (Agelenidae)], all of which are commonly found in and around human dwellings in North America [21].

# 2. Material and methods

## 2.1. Ants

*Myrmica rubra* workers (figure 1) were collected from nests at Inter River Park (49°19'10.9″ N 123°01'43.7″ W) in North Vancouver, British Columbia (BC), Canada, whereas workers of *L. niger* and *C. modoc* (figure 1) were collected from nests located on the Burnaby campus of Simon Fraser University (SFU, 49°16'33″ N 122°54'55″ W), BC. All ants were kept in jars (1–4 l) filled with soil from collection sites and were provisioned with tubes of sugar water retained with a cotton ball. To standardize the presentation of test stimuli according to weight equivalent of ants, 75 workers of each species were weighed in groups of five using a microbalance (TR-204, Denver Instrument Comp., Arvada, CO 80004, USA). Body weights (mean ± s.e.) of individual workers of *M. rubra*, *L. niger* and *C. modoc* amounted to 3.51 ± 5.56, 3.02 ± 4.44 and 43.7 ± 52.7 mg, respectively.

## 2.2. Spiders tested

All specimens of *S. grossa* (figure 1) were $F_1$ subadult offspring of mated females captured on SFU's Burnaby campus [22], whereas specimens of *L. hesperus* and *E. agrestis* (figure 1) were $F_1$ subadult offspring of mated females collected on Centennial Beach Boundary Bay Regional Park, Delta, BC (49°01'10.9″ N 123°02'32.1″ W). Spiderlings were housed singly in a Petri dish (100 × 20 mm) containing a moist cotton wick and—based on body size—were provisioned with *Drosophila* vinegar flies or *Phormia regina* blow flies once a week.

All *A. diadematus* were subadults, collected on the day of bioassays on SFU's Burnaby campus. Following bioassays, they were released into a designated non-collection zone on campus.

## 2.3. General experiments design

The effects of ant-derived deposits on aversion responses by spiders were tested in still-air, dual-choice olfactometers [8,23] kept at room temperature and a 12 L : 12 D photoperiod. Olfactometers (see fig. 1 in [8] for a photographic illustration) consisted of three circular Pyrex glass chambers (3.5 × 10 cm inner diameter (ID)) with removable glass lids linearly interconnected by glass tubes (each 2.5 × 1 cm ID). The bottoms of lateral chambers were lined with circular filter paper (Whatman, Maidstone, England). Treatment and control stimuli were assigned to lateral chambers such that the treatment stimulus was equally often presented in the left and right lateral chamber of an olfactometer to minimize any

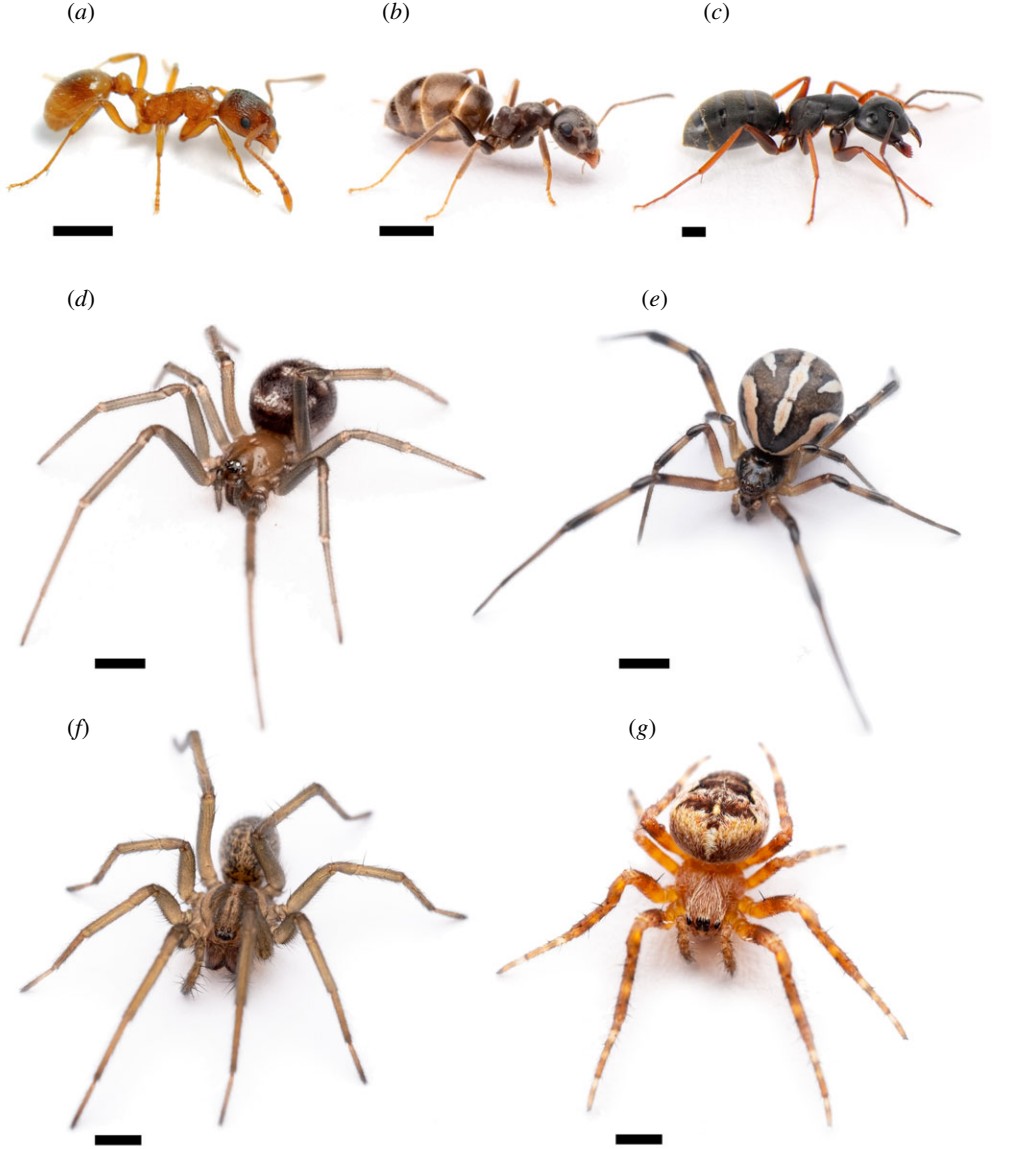

**Figure 1.** Photographs of worker ants of *Myrmica rubra* (*a*), *Lasius niger* (*b*) and *Camponotus modoc* (*c*) that were used to prepare test stimuli, and of subadult female spiders of *Steatoda grossa* (*d*), *Latrodectus hesperus* (*e*), *Eratigena agrestis* (*f*) and *Araneus diadematus* (*g*) that were tested in laboratory experiments. Bar length, 1 mm.

potential effect of side bias. To prepare a treatment stimulus, ants were placed in one lateral chamber and prevented from leaving by a wet cotton ball inserted in the glass tube interconnecting the lateral and central chamber. The wet cotton ball not only blocked the chamber exit, but also provided a source of moisture for the ants. To ensure symmetry of the experimental design, a wet cotton ball was also inserted in the glass tube interconnecting the central chamber and the second lateral chamber. As the quantity of semiochemicals deposited by ants was probably correlated with their body size or weight, equal weight equivalents of ants were used to standardize the preparation of treatment stimuli; hence, 37 *M. rubra*, 43 *L. niger* and 3 *C. modoc* were confined in the treatment chamber. After 12 h of (overnight) confinement, the ants and the cotton balls were removed. Then, a bioassay spider was introduced into the central chamber and kept in darkness for 24 h, following which its final position was scored under red light. Spiders positioned in lateral chambers were classed as responders to treatment or control stimuli, whereas those in the central chamber were recorded as non-responders. Spiders located in an interconnecting glass tube were scored as non-responders if they were closer to the central chamber than to the respective lateral chamber. All spiders were tested only once, and olfactometers were washed in detergent water (Sparkleen, Fischerbrand, Toronto, Canada) and oven-dried between replicates.

**Table 1.** List of test stimuli consisting of filter paper with chemical deposits from the ants *Myrmica rubra*, *Lasius niger* or *Camponotus modoc*, and of control stimuli invariably consisting of filter paper without chemical deposits by any ants, tested for behavioral responses of the synanthropic spiders *Steatoda grossa*, *Latrodectus hesperus*, *Eratigena agrestis* and *Araneus diadematus* in binary choice olfactometer experiments.

| Exp. no | test stimulus[a] | control stimulus | spider species bioassayed | $n$[b] |
|---|---|---|---|---|
| bioassays with *S. grossa* to test for potential side bias of olfactometers | | | | |
| 1 | no deposits | no deposits | *S. grossa* | 24 (3) |
| effect of ant species-specific chemical deposits on behavioural responses of *S. grossa* | | | | |
| 2 | 37 *M. rubra* | no deposits | *S. grossa* | 24 (3) |
| 3 | 43 *L. niger* | no deposits | *S. grossa* | 24 (1) |
| 4 | 3 *C. modoc* | no deposits | *S. grossa* | 24 (2) |
| effect of *M. rubra* chemical deposits on behavioural responses of four synanthropic spiders | | | | |
| 5 | 37 *M. rubra* | no deposits | *S. grossa* | 30 (9) |
| 6 | 37 *M. rubra* | no deposits | *L. hesperus* | 30 (11) |
| 7 | 37 *M. rubra* | no deposits | *E. agrestis* | 30 (8) |
| 8 | 37 *M. rubra* | no deposits | *A. diadematus* | 30 (11) |
| effect of *M. rubra* chemical deposit amounts on behavioural responses of *S. grossa* | | | | |
| 9 | 37 *M. rubra* | no deposits | *S. grossa* | 30 (7) |
| 10 | 111 *M. rubra* | no deposits | *S. grossa* | 30 (10) |

[a]Equal weight equivalents of ants (37 *M. rubra*, 43 *L. niger* and 3 *C. modoc*) were used to standardize the preparation of test stimuli (chemicals deposited by ants on filter paper during 12 h).
[b]$n$ = number of replicates run (number of spiders not responding in bioassays).

## 2.4. Specific experiments

Experiment 1 (table 1) was designed to reveal potential side bias associated with olfactometers. It tested the response of *S. grossa* to two control stimuli (untreated filter paper) which were presented in the lateral chambers of the olfactometer.

As there was no side bias in experiment 1 (see Results), experiments 2–4 (table 1) then tested whether semiochemicals deposited by *M. rubra* (Exp. 2), *L. niger* (Exp. 3) or *C. modoc* (Exp. 4), have a deterrent effect on *S. grossa*.

As only semiochemical deposits of *M. rubra*, but not of *L. niger* or *C. modoc*, deterred *S. grossa* (see Results), follow-up experiments 5–8 (table 1) focused on *M. rubra* semiochemicals, and tested whether they deter only *S. grossa* (Exp. 5), or also deter *L. hesperus* (Exp. 6), *E. agrestis* (Exp. 7) and *A. diadematus* (Exp. 8).

With evidence that *M. rubra* semiochemicals deter at least three spider heterogeners (see Results), experiments 9 and 10 (table 1) then tested dose-dependent effects of deterrent semiochemicals by offering *S. grossa* a choice between filter paper left untreated (control) or soiled with semiochemicals from either 37 *M. rubra* workers (Exp. 9; the same dose as in Exps. 3, 5–8) or 111 *M. rubra* workers (Exp. 10; a threefold higher dose).

## 2.5. Statistical analysis

R [24] was used to perform one-sided binominal tests to analyse data for the hypothesized repellent effect of ants on spiders in two choice experiments 1–10 [25]. Subsequently, the *p*-values of experiments were adjusted using the Benjamini–Hochberg method to account for multiple comparisons [26].

## 3. Results

When subadult *S. grossa* were offered a choice between two lateral olfactometer chambers, each containing a control stimulus (untreated filter paper), they chose the right and left chamber 11 and 10

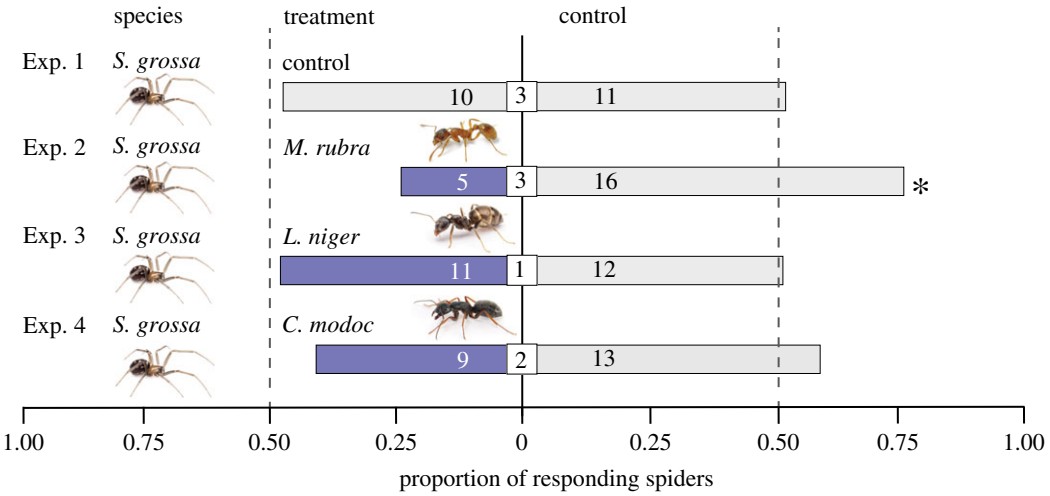

**Figure 2.** Responses of subadult *Steatoda grossa* that were given a choice in three-chamber olfactometers [23] between two test stimuli, both being untreated filter paper (Exp. 1), or one, being untreated filter paper, and the other being filter paper previously exposed to worker ants of *Myrmica rubra* (37; Exp. 2), *Lasius niger* (43; Exp. 3), or *Camponotus modoc* (3; Exp. 4). Shown within bars and square inserts are the number of spiders responding to treatment or control stimuli, and not responding to stimuli, respectively. For each experiment, an asterisk (\*) denotes a statistically significant treatment effect (one-sided binomial tests; $p < 0.05$).

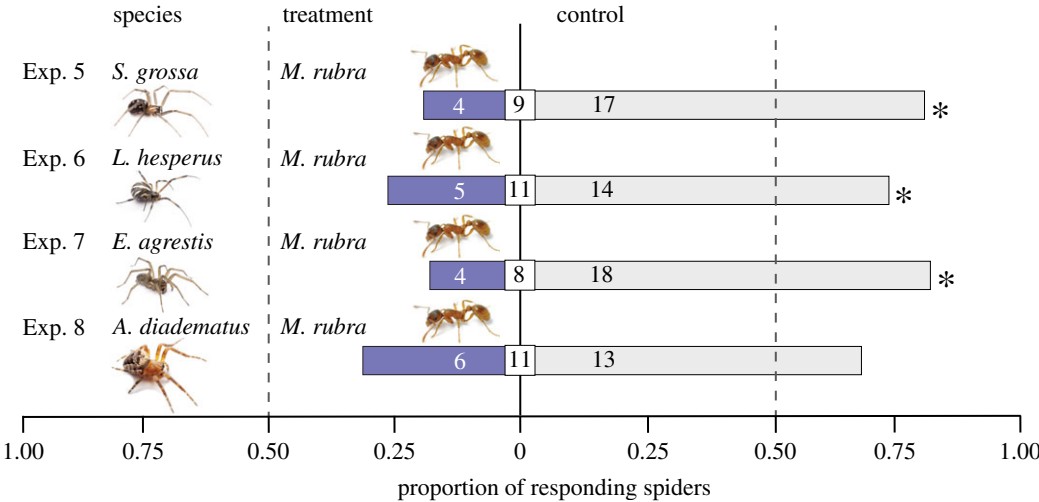

**Figure 3.** Responses of subadult *Steatoda grossa* (Exp. 5), subadult *Latrodectus hesperus* (Exp. 6), subadult *Eratigena agrestis* (Exp. 7), and subadult *Araneus diadematus* (Exp. 8) that were given a choice in three-chamber olfactometers [23] between two test stimuli, one being untreated filter paper and the other being filter paper previously exposed to 37 worker ants of *Myrmica rubra*. Shown within bars and square inserts are the number of spiders responding to treatment or control stimuli, and not responding to stimuli, respectively. For each experiment, an asterisk (\*) denotes a statistically significant treatment effect (one-sided binomial tests; $p < 0.05$).

times, respectively, revealing no evidence for a side bias ($p = 0.50$; Exp. 1, figure 2). Semiochemicals deposited by *M. rubra* had a significant deterrent effect on *S. grossa* (Exp. 2: spiders in treatment chamber (5) versus spiders in control chamber (16), $p = 0.004$; figure 2). By contrast, semiochemicals deposited by *L. niger* (Exp. 3) or *C. modoc* (Exp. 4) failed to deter *S. grossa* (Exp. 3: 11 versus 12; $p = 0.50$; Exp. 4: 9 versus 13, $p = 0.393$; figure 2). In parallel experiments 5–8, semiochemicals deposited by *M. rubra* had a significant deterrent effect on *S. grossa* (Exp. 5: 4 versus 17; $p = 0.007$, figure 3), *L. hesperus* (Exp. 6: 5 versus 14; $p = 0.042$, figure 3) and *E. agrestis* (Exp. 7: 4 versus 18; $p = 0.007$, figure 3), but not on *A. diadematus* (Exp. 8: 6 versus 13; $p = 0.084$, figure 3). There was a dose-dependent effect of the amount of semiochemicals deposited by *M. rubra* on behavioural responses of *S. grossa*. The amount of semiochemicals deposited by 111 *M. rubra* had a deterrent effect on *S. grossa* (Exp. 10: 5 versus 15, $p = 0.041$, figure 4) but the aversion effect caused by deposits of only 37 *M. rubra* was not statistically significant in this particular experiment (Exp. 9: 8 versus 15; $p = 0.105$, figure 4).

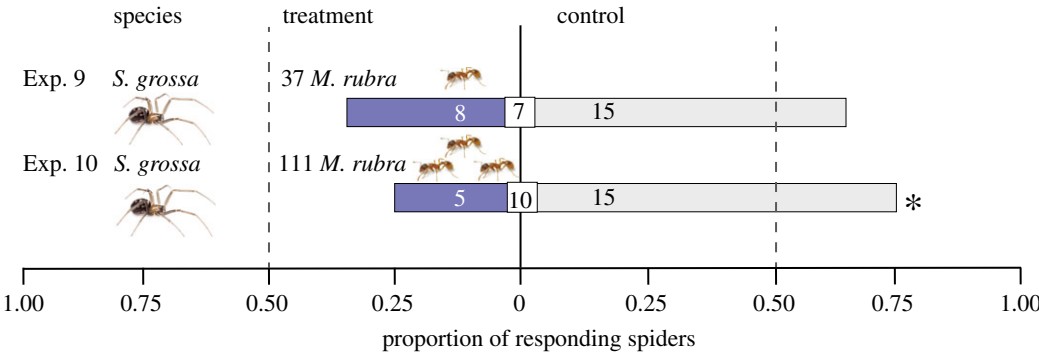

**Figure 4.** Responses of subadult *Steatoda grossa* that were given a choice in three-chamber olfactometers [23] between two test stimuli, one being untreated filter paper and the other being filter paper previously exposed to 37 or 111 worker ants of *Myrmica rubra* (Exps. 9 and 10, respectively). Shown within bars and square inserts are the number of spiders responding to treatment or control stimuli, and not responding to stimuli, respectively. For each experiment, an asterisk (*) denotes a statistically significant treatment effect (one-sided binomial tests; $p < 0.05$).

## 4. Discussion

Our data support the conclusion that semiochemical deposits of *M. rubra* worker ants have a significant deterrent effect on three spider species (*S. grossa*, *L. hesperus*, *E. agrestis*) and that they express a moderate deterrent effect on a fourth spider species tested in our study, the cross spider *A. diadematus*. Conversely, semiochemical deposits of *L. niger* and *C. modoc* worker ants failed to induce a discernible behaviour-modifying effect on the spiders tested.

Our findings that semiochemical deposits of *M. rubra* worker ants, but not of *L. niger* or *C. modoc* worker ants, prompted aversion responses by *S. grossa* have multiple potential explanations, such as the specifics of the experimental design, contrasting life-history traits of ants, and niche overlap, or not, between ants and spiders.

As part of the experimental design to prepare ant semiochemical deposits as test stimuli for spiders, we selected diverse taxonomic species of ants that greatly varied in body size and weight. Assuming that larger ants deposit greater amounts of semiochemicals, we standardized the amount of deposits between experiments by testing equal weight equivalents of ants, using 37, 43 and 3 worker ants of *M. rubra*, *L. niger* and *C. modoc*, respectively, to generate a test stimulus. However, contrary to our assumption, the body weight of ants and the amount of semiochemical deposits may not be positively correlated, and equal numbers, rather than equal weights, of *M. rubra*, *L. niger* and C. *modoc* worker ants may have been required to generate standardized test stimuli. Alternatively, the semiochemicals deposited by *M. rubra* may have significantly greater potency as spider deterrents than those of *L. niger* and *C. modoc*. Worker ants of *M. rubra* are omnivorous scavengers and prey on many invertebrates [27]. In their invaded North American range, populations of *M. rubra* occur in extremely high densities and appear more aggressive than their counterparts in Europe. These characteristics, coupled with efficient foraging and aggressive nest defence, have enabled *M. rubra* to outcompete native ants and lower the arthropod biodiversity in invaded communities [28]. It is conceivable then, that over evolutionary time arthropod community members, including spiders which may fall prey to *M. rubra*, have learned to respond to semiochemical cues of *M. rubra* and to settle in (micro) habitats void of *M. rubra*. If so, this would provide ecological rationale for our data showing that semiochemical deposits of *M. rubra* have deterrent effects on *S. grossa*, *L. hesperus* and *E. agrestis*.

Insufficient niche overlap between *M. rubra* and *A. diadematus*, and thus a lack of opportunity to learn each other's semiochemical signals or cues, may explain why semiochemical deposits of *M. rubra* had only a weak (and statistically not significant) deterrent effect on *A. diadematus*. As orb-weavers, *A. diadematus* females build their webs above ground [29], physically well separated from the subterranean colonies of *M. rubra*. Females of *S. grossa*, *L. hesperus* and *E. agrestis*, in contrast, build their three-dimensional cobwebs near ground level [29] with greater likelihood of frequent encounters with foraging *M. rubra* workers.

The identity of the deterrent semiochemical(s) deposited by *M. rubra* workers remains unknown. Communication signals such as trail or alarm pheromones [30–34] are least likely to be the deterrent(s) because the sets of 37 *M. rubra* workers used to generate test stimuli in the confines of

olfactometers had no immediately obvious incentive to release pheromone and coordinate activities. Yet, signalling in ants is complex and we are just beginning to grasp that complexity. While the functional role of most exocrine glands in *M. rubra* [35] is still unknown, any gland may have released the semiochemical(s) that prompted the deterrent effect on spiders. Alternatively, the semiochemicals are not released from glands but originate from the ants' body surface.

Irrespectively, the rather remarkable deterrence of *M. rubra* semiochemical deposits against *S. grossa*, *L. hesperus* and *E. agrestis* warrant the identification of these deterrents through proven-effective techniques in arthropod chemical ecology [36]. Once identified, the origin of these deterrents could be traced to a specific exocrine gland and/or the body surface of ants. Moreover, synthetic replica of these deterrents could be developed, together with concurrently known spider deterrents [9], for earth-friendly manipulation of synanthropic spiders.

Data accessibility. All data are presented in the manuscript and in the electronic supplementary material [37].

Authors' contributions. Conceptualization and methodology: A.F. and G.G.; formal analysis and investigation: T.D., A.F. and Y.L.; writing original draft: Y.L., T.D. and A.F.; draft review and editing: G.G. and A.F.; all authors reviewed and approved the final draft; funding acquisition, resources and supervision: G.G.

Competing interests. The authors declare no conflicts of interest.

Funding. The research was supported by a Natural Sciences and Engineering Research Council of Canada (NSERC) – Alexander Graham Bell Scholarship to A.F., an NSERC – Undergraduate Student Research Award to Y.L., and by an NSERC – Industrial Research Chair to G.G., with BASF Canada Inc. and Scotts Canada Ltd as the industrial sponsors.

Acknowledgements. We thank Jaime Chalissery for helpful input, Sharon Oliver for some word processing and comments, and the Associate Editor Dr Kimberley Mathot and two anonymous reviewers for constructive comments.

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
