## [Peer Review File · Royal Society Open Science]

Review History

RSOS-210279.R0 (Original submission)

Review form: Reviewer 1

Is the manuscript scientifically sound in its present form?

Yes

Are the interpretations and conclusions justified by the results?

Yes

Is the language acceptable?

Yes

Do you have any ethical concerns with this paper?

No

Have you any concerns about statistical analyses in this paper?

No

Recommendation?

Accept with minor revision (please list in comments)

Comments to the Author(s)

Review of RSOS 210279

Overall, a well-conducted study with a logical flow of the string of experiments. The figures are exceptionally self-explanatory.

General comments

1- I don't see why some common names are capitalized. I would think it should be "black garden ants, hobo spiders, false black widow". Capitalization would seem to be used for more important proper words like "European fire ants".

2- although olfactometers are standard, some vary in construction and I think the paper would benefit from an image of the apparatus. I had trouble visualizing the set up.

3- the accepted common name for *A. diadematus* is the cross spider.

Specific comments

line 40: delete "a"

line 77: spell out "British Columbia"

line 160: I think the order of numbers should be "37, 43 and 3"

line 177: I would change "learn" to "react to". I seriously doubt that learning is involved. It seems more likely that the spiders have an innate response to semiochemicals in a general sense.

citations 32 and 34: "*Myrmica*" should be in italics, Check elsewhere for others

Figure 3 legend: "*Latrodectus*" is misspelled

Review form: Reviewer 2

Is the manuscript scientifically sound in its present form?

No

Are the interpretations and conclusions justified by the results?

No

Is the language acceptable?

Yes

Do you have any ethical concerns with this paper?

No

Have you any concerns about statistical analyses in this paper?

No

Recommendation?

Major revision is needed (please make suggestions in comments)

Comments to the Author(s)

The manuscript by Fisher et al. presented data on the deterrence of spiders by European ant species. The manuscript is well written in its current form and the methods used are sound as far as bioassays are concerned. My major concerns with the manuscript is that it only presents a proof of concept indicting olfactory cues in this relationship. But the semiochemicals were not identified. It is critical to identify the chemicals deposited and at least indicate the profiles which would possibly identify qualitative constituents of these and present reasons why the deterrent effects differ between the ant species. Identifying, elucidating the individual components and using these individually or as mixtures in bioassays to determine those that are responsible for the observed responses would prove their roles.

Minor comment: Describing the experiments using respective numbers in the text makes it difficult to follow. I suggest the different experiments be listed and the combinations in a supplement (if this cannot be in the text).

Decision letter (RSOS-210279.R0)

Dear Mr Fischer

On behalf of the Editors, we are pleased to inform you that your Manuscript RSOS-210279 "Know your foe -Synanthropic spiders are deterred by semiochemicals of European fire ants" has been accepted for publication in Royal Society Open Science subject to minor revision in accordance with the referees' reports. Please find the referees' comments along with any feedback from the Editors below my signature.

Please submit your revised manuscript and required files (see below) no later than 7 days from today's (ie 07-Apr-2021) date. Note: the ScholarOne system will 'lock' if submission of the revision is attempted 7 or more days after the deadline. If you do not think you will be able to meet this deadline please contact the editorial office immediately.

Kind regards,
Royal Society Open Science Editorial Office
Royal Society Open Science

on behalf of Dr Kimberley Mathot (Associate Editor) and Pete Smith (Subject Editor)
openscience@royalsociety.org

Associate Editor Comments to Author (Dr Kimberley Mathot):

Associate Editor: 1

Comments to the Author:

The manuscript has now been reviewed by two experts in the field. Overall, both referees felt the experiments were well designed and clearly described, and that the manuscript could be published, pending revisions. My own assessment is that the required revisions are relatively straightforward, and I am recommending the manuscript be accepted pending minor revisions.

Specifically, referee 1 asks for a more detailed description of the olfactometer setup, potentially including a schematic. They also offer other very minor comments/corrections, which should be implemented in the revised version. Referee 2 had two main concerns. First, that referring to the experiments as a series of numbers was difficult to follow. I agree, and would recommend that in addition to the numbers, experiments be given short, descriptive titles so that readers do not have to memorize which specific experimental conditions each number refers to. Referee 2's second concern is related to the fact that while the study indicates that olfactory cues are used, the specific semiochemicals are not identified. While true, I don't see this as a critical flaw, but rather a limitation of the study. I feel that the authors have acknowledged and discussed this shortcoming sufficiently in the discussion (e.g., lines 183-197).

Reviewer comments to Author:

Reviewer: 1

Comments to the Author(s)

Review of RSOS 210279

Overall, a well-conducted study with a logical flow of the string of experiments. The figures are exceptionally self-explanatory.

General comments

1- I don't see why some common names are capitalized. I would think it should be "black garden ants, hobo spiders, false black widow". Capitalization would seem to be used for more important proper words like "European fire ants".

2- although olfactometers are standard, some vary in construction and I think the paper would benefit from an image of the apparatus. I had trouble visualizing the set up.

3- the accepted common name for *A. diadematus* is the cross spider.

Specific comments

line 40: delete "a"

line 77: spell out "British Columbia"

line 160: I think the order of numbers should be "37, 43 and 3"

line 177: I would change "learn" to "react to". I seriously doubt that learning is involved. It seems more likely that the spiders have an innate response to semiochemicals in a general sense.

citations 32 and 34: "Myrmica" should be in italics, Check elsewhere for others

Figure 3 legend: "Latrodectus" is misspelled

Reviewer: 2

Comments to the Author(s)

The manuscript by Fisher et al. presented data on the deterrence of spiders by European ant species. The manuscript is well written in its current form and the methods used are sound as far as bioassays are concerned. My major concerns with the manuscript is that it only presents a proof of concept indicting olfactory cues in this relationship. But the semiochemicals were not identified. It is critical to identify the chemicals deposited and at least indicate the profiles which would possibly identify qualitative constituents of these and present reasons why the deterrent effects differ between the ant species. Identifying, elucidating the individual components and using these individually or as mixtures in bioassays to determine those that are responsible for the observed responses would prove their roles.

Minor comment: Describing the experiments using respective numbers in the text makes it difficult to follow. I suggest the different experiments be listed and the combinations in a supplement (if this cannot be in the text).

===PREPARING YOUR MANUSCRIPT===

===PREPARING YOUR REVISION IN SCHOLARONE===

Author's Response to Decision Letter for (RSOS-210279.R0)

See Appendix A.

Decision letter (RSOS-210279.R1)

Dear Mr Fischer,

It is a pleasure to accept your manuscript entitled "Know your foe –Synanthropic spiders are deterred by semiochemicals of European fire ants" in its current form for publication in Royal Society Open Science. The comments from the Editors are included at the foot of this letter.

You can expect to receive a proof of your article in the near future. Please contact the editorial office (openscience@royalsociety.org) and the production office (openscience_proofs@royalsociety.org) to let us know if you are likely to be away from e-mail contact – if you are going to be away, please nominate a co-author (if available) to manage the proofing process, and ensure they are copied into your email to the journal.

on behalf of Dr Kimberley Mathot (Associate Editor) and Pete Smith (Subject Editor)
openscience@royalsociety.org

Associate Editor Comments to Author (Dr Kimberley Mathot):

Thank you for submitting your revised manuscript and your careful attention to detail. I am happy to recommend your paper for publication.

Appendix A

Responses (R) to Dr. Mathot's comments and to comments provided by two anonymous reviewers

Associate Editor Comments to Author (Dr Kimberley Mathot):

Comments to the Author:

1. The manuscript has now been reviewed by two experts in the field. Overall, both referees felt the experiments were well designed and clearly described, and that the manuscript could be published, pending revisions. My own assessment is that the required revisions are relatively straightforward, and I am recommending the manuscript be accepted pending minor revisions.

R1: We appreciate the positive assessment of our manuscript by you, Dr. Mathot, and by the anonymous reviewers, and we appreciate the opportunity to improve the manuscript.

2. Specifically, referee 1 asks for a more detailed description of the olfactometer setup, potentially including a schematic. They also offer other very minor comments/corrections, which should be implemented in the revised version.

R2: While this manuscript was in review, a photographic illustration of the very same olfactometer was published in Fischer et al. (2021) (reference [8] in this manuscript). It is conventional practice not to repeatedly show the same photograph in related articles, but to refer to the previous publication that first reported the olfactometer. To address this comment, we have revised the text, as follows (line 85 in the revised MS): 'Olfactometers ('see Fig. 1 in [8]) for a photographic illustration) consisted of....' We have carefully reviewed the description of the olfactometer in the main text and found that all details are properly described.

3. Referee 2 had two main concerns. First, that referring to the experiments as a series of numbers was difficult to follow. I agree, and would recommend that in addition to the numbers, experiments be given short, descriptive titles so that readers do not have to memorize which specific experimental conditions each number refers to.

R3: We have tried to add a descriptive title to each experiment but – as a result – the text became 'choppy' and awkward to read. Therefore, we have accepted the alternative suggestion made by Reviewer #2 that '*different experiments be listed*'. We have prepared a Table that (i) summarizes all experiments, (ii) provides descriptive titles for single or sets of experiments, and (iii) reports the stimuli that were prepared and the species of spiders that were tested. We refer to this Table in 'Methods' as a quick guide as to which stimuli were tested and the underlying rationale.

4. Referee 2's second concern is related to the fact that while the study indicates that olfactory cues are used, the specific semiochemicals are not identified. While true, I don't see this as a critical flaw, but rather a limitation of the study. I feel that the authors have acknowledged and discussed this shortcoming sufficiently in the discussion (e.g., lines 183-197).

R4: We agree with your judgement.

Reviewer: 1

5. Overall, a well-conducted study with a logical flow of the string of experiments. The figures are exceptionally self-explanatory.

R5: We thank Reviewer 1 for the kind remarks, particularly about the figures.

General comments

6. 1- I don't see why some common names are capitalized. I would think it should be "black garden ants, hobo spiders, false black widow". Capitalization would seem to be used for more important proper words like "European fire ants".

R6: Revised accordingly.

7. 2- although olfactometers are standard, some vary in construction and I think the paper would benefit from an image of the apparatus. I had trouble visualizing the set up.

R7: Please see R2.

8. 3- the accepted common name for *A. diadematus* is the cross spider.

R8: Revised accordingly.

Specific comments

9. line 40: delete "a"

R9: Revised accordingly.

10. line 77: spell out "British Columbia"

R10: Revised accordingly.

11. line 160: I think the order of numbers should be "37, 43 and 3"

R11: Thank you for catching this. Revised accordingly

12. line 177: I would change "learn" to "react to". I seriously doubt that learning is involved. It seems more likely that the spiders have an innate response to semiochemicals in a general sense.

R12: We contend that some kind of learning must have taken place in evolutionary times. For example, why would *Latrodectus* have an innate avoidance response to European fire ants which are only recent adventives in North America? We have revised to text to read (line 168): It is conceivable then that over evolutionary time arthropod community members, including spiders which may fall prey to *M. rubra*, have learned to respond to semiochemical cues of *M. rubra* and to settle in (micro) habitats void of *M. rubra*.

13. citations 32 and 34: "Myrmica" should be in italics, Check elsewhere for others

R13: Revised accordingly.

14. Figure 3 legend: "Latrodectus" is misspelled

R14: Thank you for pointing out this typo! Corrected accordingly.

Reviewer: 2

Comments to the Author(s)

15. The manuscript by Fisher et al. presented data on the deterrence of spiders by European ant species. The manuscript is well written in its current form and the methods used are sound as far as bioassays are concerned.

R15: Thank you for the overall positive assessment.

16. My major concerns with the manuscript is that it only presents a proof of concept indicting olfactory cues in this relationship. But the semiochemicals were not identified. It is critical to identify the chemicals deposited and at least indicate the profiles which would possibly identify qualitative constituents of these and present reasons why the deterrent effects differ between the ant species. Identifying, elucidating the individual components and using these individually or as mixtures in bioassays to determine those that are responsible for the observed responses would prove their roles.

R16: Yes, the semiochemicals were not identified. However, based on our extensive experience in the identification of semiochemicals this is a major research project that is beyond the scope of this study. Please also see R4.

17. Minor comment: Describing the experiments using respective numbers in the text makes it difficult to follow. I suggest the different experiments be listed and the combinations in a supplement (if this cannot be in the text).

R17: Please see R3.